# Wide-Area and Real-Time Object Search System of UAV

Xianjiang Li [1], Boyong He [1], Kaiwen Ding [1], Weijie Guo [2], Bo Huang [1] and Liaoni Wu [1,*]

1    School of Aerospace Engineering, Xiamen University, Xiamen 361102, China;
     lixianjiang@stu.xmu.edu.cn (X.L.); heboyong0220@stu.xmu.edu.cn (B.H.);
     dingkaiwen@stu.xmu.edu.cn (K.D.); huangbo@stu.xmu.edu.cn (B.H.)
2    School of Informatics, Xiamen University, Xiamen 361005, China; ccgwj@stu.xmu.edu.cn
*    Correspondence: wuliaoni@xmu.edu.cn

**Abstract:** The method of collecting aerial images or videos by unmanned aerial vehicles (UAVs) for object search has the advantages of high flexibility and low cost, and has been widely used in various fields, such as pipeline inspection, disaster rescue, and forest fire prevention. However, in the case of object search in a wide area, the scanning efficiency and real-time performance of UAV are often difficult to satisfy at the same time, which may lead to missing the best time to perform the task. In this paper, we design a wide-area and real-time object search system of UAV based on deep learning for this problem. The system first solves the problem of area scanning efficiency by controlling the high-resolution camera in order to collect aerial images with a large field of view. For real-time requirements, we adopted three strategies to accelerate the system, as follows: design a parallel system, simplify the object detection algorithm, and use TensorRT on the edge device to optimize the object detection model. We selected the NVIDIA Jetson AGX Xavier edge device as the central processor and verified the feasibility and practicability of the system through the actual application of suspicious vehicle search in the grazing area of the prairie. Experiments have proved that the parallel design of the system can effectively meet the real-time requirements. For the most time-consuming image object detection link, with a slight loss of precision, most algorithms can reach the 400% inference speed of the benchmark in total, after algorithm simplification, and corresponding model's deployment by TensorRT.

**Keywords:** unmanned aerial vehicles (UAVs); wide-area and real-time object search; aerial image; object detection

## 1. Introduction

The development of technology in recent years has promoted the popularization of UAV in object search tasks. For instance, many teams use the UAV to carry out ship detection [1,2], object tracking [3], and emergency search and rescue [4,5] in the ocean environment. In the field of agriculture, the UAV has a wide range of applications, such as weed recognition [6,7], pest detection [8], and precision pesticide spraying [9], which effectively promotes the intelligentization of agriculture. Forests are also one of the typical scenarios of object search, and common applications include fire monitoring and prevention [10,11], ecological protection [12,13], etc.

In most of these tasks, object search is mainly performed by collecting images or videos and processing them through a search algorithm. For the real-time object search of UAV aerial images or videos, the traditional feature extraction algorithms have high computational complexity and are time consuming, which cannot meet the real-time requirements. In 2012, with the outstanding performance of the AlexNet [14] deep learning model based on convolutional neural network (CNN) in the ImageNet [15] image classification competition, people began to realize the powerful capabilities of CNN in image feature extraction and analysis. In the object-detection branch, benefitting from the continuous exploration of deep learning technology in recent years, rich and diverse algorithms have been proposed,

including RCNN series [16–18], SSD [19], YOLO series [20–23], and excellent practical application effects have been achieved. The commonly used object detection algorithms can be roughly divided into two categories. The first category is a regression-based single-stage algorithm, such as SSD [19], YOLOv4 [23], and RetinaNet [24]; another category is a two-stage algorithm based on region proposal and regression, such as Faster RCNN [18] and Cascade RCNN [25]. Generally, the single-stage detection algorithms are efficient, but the precision is relatively low, while the two-stage algorithms are inefficient but have higher detection precision. Recently, anchor-free type algorithms, such as FCOS [26] and ATSS [27], have also emerged in the field of object detection, which has reduced the amount of calculation that uses anchors. The classification of commonly used object detection algorithms can be seen in Figure 1.

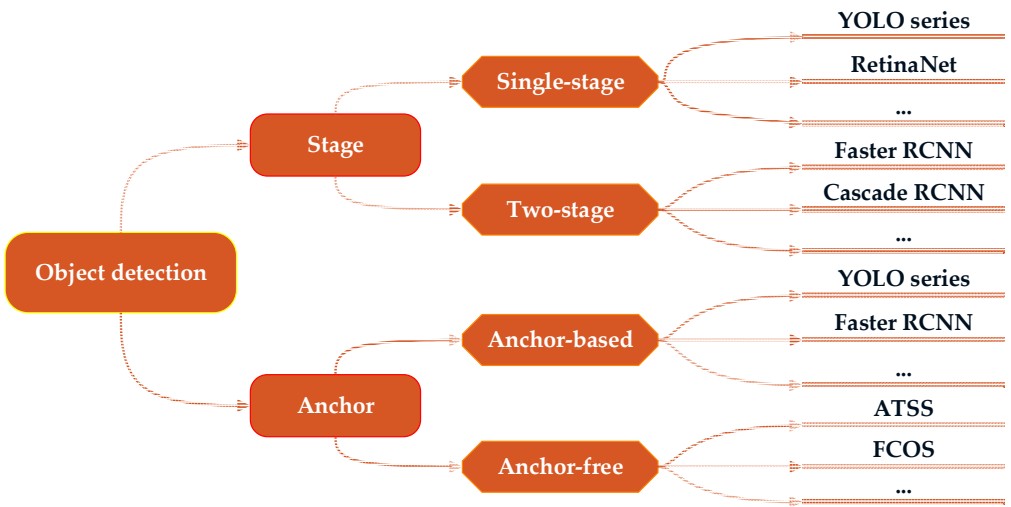

**Figure 1.** Classification of commonly used object detection algorithms.

When performing tasks that require UAV to search for objects in a wide area, two main issues of "area scanning efficiency" and "real-time detection" need to be considered. In terms of area scanning efficiency, an effective and easy-to-implement method is to use the UAV equipped with high-resolution cameras to take images of the ground surface at mid-to-high altitude. Figure 2 displays the characteristics of the images collected in this way. As shown in the figure, the aerial images collected at mid-to-high altitude have the characteristics of a large field of view and small objects. When the UAV's flight altitude relative to the ground is 900 m, and the camera is 42.4 megapixels ($7952 \times 5304$ pixels), while the ground vehicle in the image only accounts for 0.0075% (about $80 \times 40$ pixels) of all of the pixels in the entire image. Since the cameras are just mounted on the UAV, the image data can only be taken out after the UAV has landed and then processed and filtered through the object detection algorithm. Undoubtedly this method takes much post-processing time and costs more to perform. In some time-sensitive task situations, it may also lead to missing the best time for the object search.

For the real-time detection issue, there have been many related kinds of research on the practical application of UAV object search. Some research mainly focus on simplifying a specific object detection algorithm, making it faster and more suitable for mobile devices with limited computing power. Zhan W. et al. [28] improved the YOLOv5 object detection algorithm from four aspects in order to achieve real-time detection of small objects, as follows: by redesigning the anchor size, adding attention module to the backbone, using CIOU loss function, and adding the P2 feature level [29] proposes ShuffleDet based on ShuffleNet [30], and a modified variant of SSD [19] to realize real-time vehicle detection by UAV. While improving the algorithm, there is also a part of research that describes the hardware composition and the workflow of the real-time object search system of UAV [31], which describes a real-time survivor detection system with a pruned object

detection algorithm in a UAV, proposed in order to reduce the loss of lives caused by natural disasters. In [32], Chen, L. et al. collected video stream and executed real-time object detection by carrying a camera, then controlling the UAV to perform corresponding actions. Other related research includes [33–38].

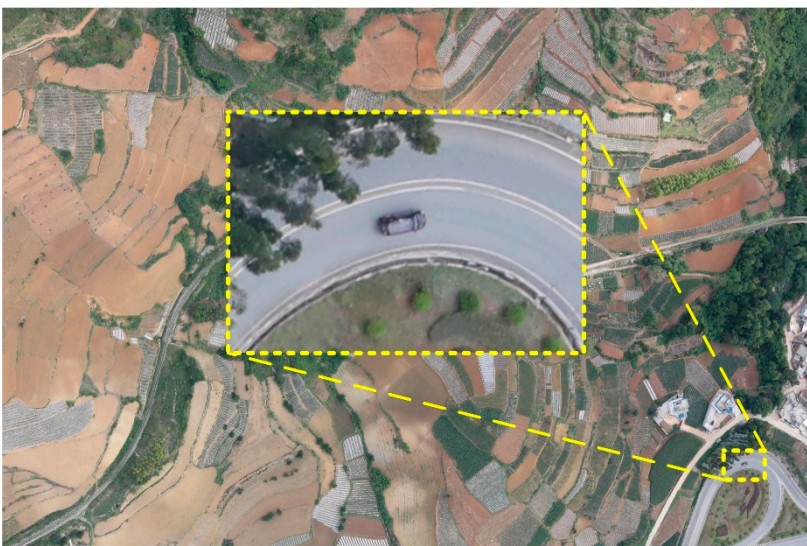

**Figure 2.** Small object in aerial image with a large field of view. The pixels of the vehicle in this image only account for 0.0075% of the whole image pixels. Although it is very efficient to search for objects by collecting large field view aerial images, it poses a challenge to real-time requirements.

Most of the UAV real-time object search systems that are involved in previous research [31,32,36–38] are more suitable for low-altitude and short-endurance flight tasks, and the equipped equipment is often a small-resolution camera for taking images or videos; there is little research on real-time systems when the UAV flies at mid-to-high altitude. While many UAVs can be used in parallel to achieve wide-area coverage, this requires the addition of multiple UAVs and ground crews, resulting in a significant increase in the operating costs.

The advantage of searching by collecting aerial images is that a large area is scanned at one time, which is highly efficient, but it is a challenge to the real-time performance of the system. In this paper, we have carried out the following points in order to design a system that satisfies both area scanning efficiency and real-time detection performance:

- We designed a wide-area and real-time object search system of UAV using a high-resolution camera and NVIDIA Jetson AGX Xavier embedded edge device, and verified the system's feasibility in an actual task. With the help of the parallel computing capabilities of the edge device, the system adopts the idea of parallel modular design to improve the system's performance. Based on this design, the system can also flexibly adapt different types of cameras and objection detection algorithms according to the task requirements, thereby increasing the versatility and configurability of the system;

- Considering the diversity of the object detection algorithm selection caused by the diversity of tasks, we adopted some general strategies to simplify the commonly used object detection algorithms in order to improve the inference speed instead of optimizing for a specific algorithm. Moreover, the TensorRT deployment method is used to further accelerate the object detection model on the NVIDIA Jetson AGX Xavier edge device.

Due to the broad coverage of a single aerial image, accurate positioning of the object in the image is essential. Otherwise, the object positioning will significantly deviate from the actual location. In many studies, such as [31,38], there are a lack of descriptions for object positioning, because the object occupies a large proportion of the image frame at a limited

altitude, in which it can be approximately considered that the object location is equivalent to the location of the UAV. Ref. [37] mentioned the study of velocity estimation of the tracked object, but also no precise coordinate localization. This paper will also introduce the detailed calculation method to perfect the whole system.

For the selection of object detection algorithms, this paper chooses the mainstream algorithms that are commonly used in engineering as the benchmark, which have higher detection precision [32] and chose the lightweight YOLOv3-tiny algorithm, which is fast, but the detection precision is relatively low and will degrade the system's overall performance. After appropriate acceleration of the selected algorithms, their detection speed has significantly improved at the expense of a small amount of precision, allowing them to meet the real-time requirements in the system.

## 2. System Design and Optimization

The wide-area and real-time object search system comprises software and various hardware, as depicted in Figure 3. Simply put, if a wide-area object search task needs to be performed, the UAV will use the camera to scan the area to be searched at a mid-to-high altitude by taking images. The captured image will immediately be read into the onboard edge device (we chose the Jetson AGX Xavier, which will be introduced later), which applies the objection detection algorithm for object search. Then, the detected object image and corresponding positioning information will be transmitted to the ground control station (GCS), in real-time, for confirmation. After determining the suspicious object, the decision will be made as to whether to go to the scene for further inspection, according to the situation.

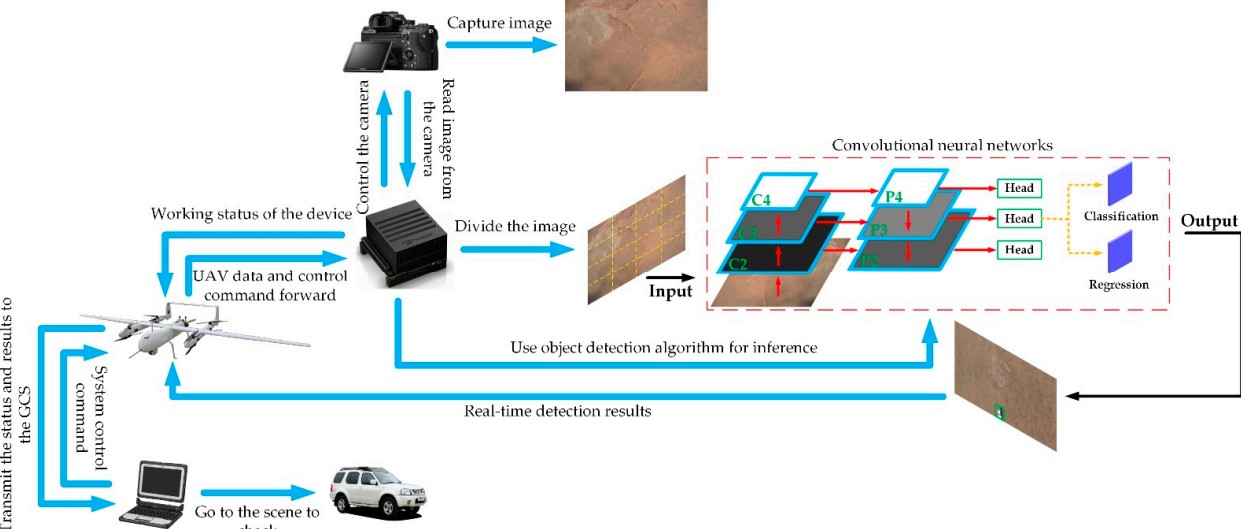

**Figure 3.** Wide-area and real-time object search system of UAV. It is a system composed of software and various hardware.

### 2.1. The Main Hardware of System

Most of the hardware that is used in this system is shelf products, which can be easily purchased without special customization, which saves the construction cost of the entire system, to a certain extent.

#### 2.1.1. Jetson AGX Xavier

Jetson AGX Xavier is an embedded AI edge device produced by NVIDIA. The device is equipped with a Xavier processor with eight 64-bit ARM architecture CPUs, while the GPU uses NVIDIA Volta architecture with 512 NVIDIA CUDA cores and 64 Tensor cores. This device provides a peak computing power of up to 32 TOPS and a high-speed I/O transmission of 750 Gbps. Compared with the previous generation Jetson TX2, the

performance has improved 20 times. In order to ensure the interaction with external devices in multiple ways simultaneously, the device integrates peripheral interfaces, such as PCIE, Gigabit Ethernet, UART serial port, type-c, and HDMI. Table 1 lists the performance comparison of some processors. We can see that Jetson AGX Xavier has the highest AI performance, and the memory also reaches 32 GB, which is very beneficial for model execution and large-volume image processing. Whether or not TensorRT can be used is also an important indicator because it can accelerate the model and achieve a high addition to the detection speed.

**Table 1.** Performance comparison of some processors. TFLOPS stands for tera floating point operations per second, and TOPS stand for tera operations per second.

| Device Type | Memory Size | Max Power | AI Performance | TensorRT |
|---|---|---|---|---|
| Jetson AGX Xavier | 32 GB | 30 W | 32 TOPS | Support |
| Jetson Xavier NX | 8 GB | 20 W | 21 TOPS | Support |
| Jetson TX2 | 8 GB | 15 W | 1.33 TFLOPS | Support |
| Atlas 500 | 8 GB | 40 W | 22 TOPS | / |

While having computing power comparable to a workstation, Jetson AGX Xavier also has a small size of $100 \times 100 \times 87$ mm$^3$ (see Figure 4 for the appearance), which is very suitable for deployment on mobile platforms, such as UAVs, with limited carrying capacity. In our system, the device is mainly responsible for controlling the camera, communicating with the UAV, and detecting and positioning the suspected objects in the image.

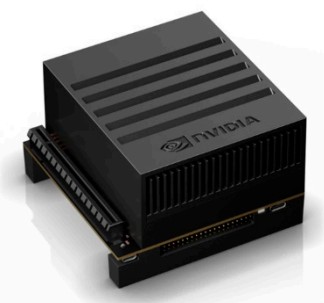 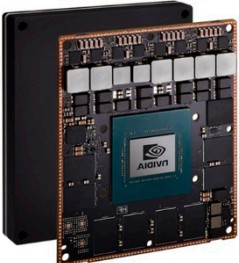

**Figure 4.** NVIDIA Jetson AGX Xavier edge device, which was selected as the system's central controller.

### 2.1.2. Camera

There are many types of cameras to choose from, but this paper mainly discusses high-resolution visible-light cameras. We have tested several common cameras on the market, and their appearances are shown in Figure 5. The comparison of these cameras is shown in Table 2.

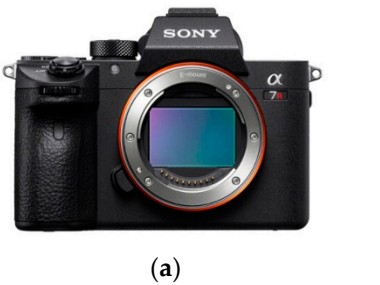

(**a**)

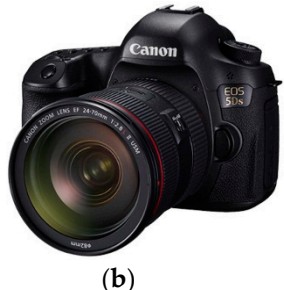

(**b**)

**Figure 5.** (**a**) Sony A7R series cameras. They have a similar appearance; (**b**) The appearance of a Canon 5DS camera.

**Table 2.** Comparison of working performance of different cameras. The "Capture Interval" represents the average time from when the camera is triggered to when the image is transmitted to the edge device.

| Camera Type | Camera Weight | Megapixels | Image Volume | Capture Interval |
|---|---|---|---|---|
| Sony A7R2 | 625 g | 42.4 | 30.0 MB | 3.6 s |
| Sony A7R3 | 657 g | 42.4 | 30.0 MB | 2.4 s |
| Sony A7R4 | 665 g | 61.0 | 47.5 MB | 2.8 s |
| Canon 5DS | 930 g | 50.6 | 35.0 MB | 2.5 s |

The most significant difference between aerial image capture and video capture is that there is a longer acquisition time interval between each aerial image. An acquisition delay of 2 to 4 s is acceptable in the actual application, as long as the images can cover the entire area that is to be searched. It can be seen from Figure 6 that the images taken along the flying direction of the UAV can cover the ground completely, while ensuring some overlap. Since the attitude of the UAV changes in real-time during the flight, there are some misalignments in the images, which can later be corrected by the positioning algorithm. We selected Sony A7R2 to perform the system verification in the actual task. From the data of Table 2, it seems that Sony A7R2 is not a good choice, but in fact, it already meets the working requirements, and our experiments will also prove that, within a certain range, the normal work of the system has nothing to do with the choice of camera, but the better the camera, the better the performance of the system.

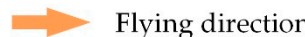 Flying direction of the UAV

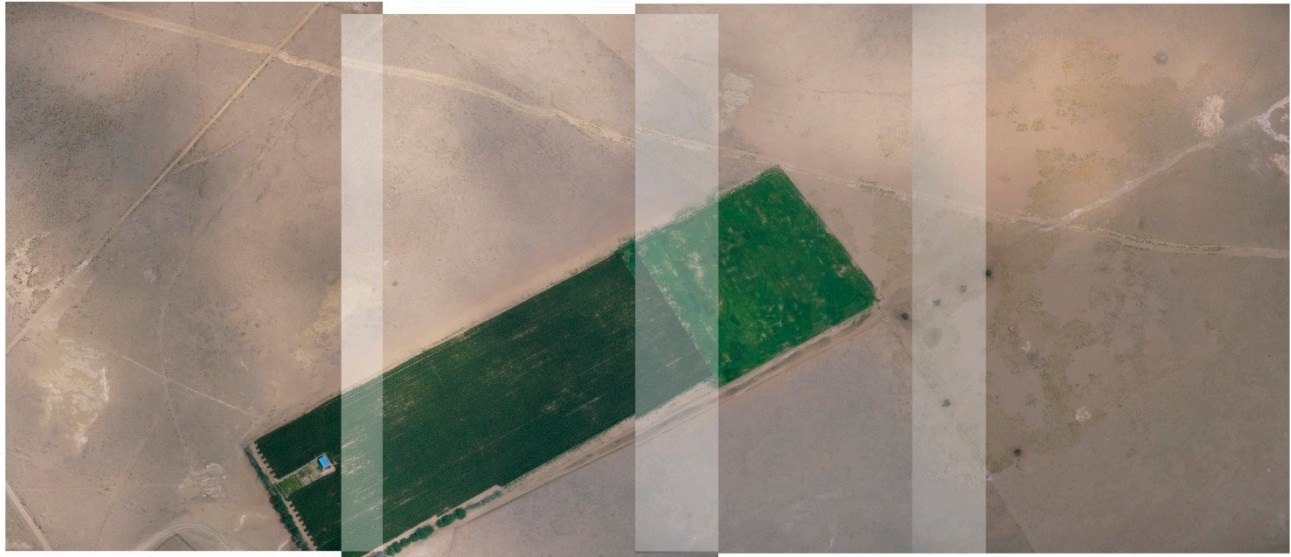

**Figure 6.** Aerial images cover the ground with some overlap.

### 2.1.3. UAV

A strong endurance capability is essential for an UAV that conducts wide-area object searches. In addition, the UAV also needs to have an interface to communicate with the object search system. In order to meet these requirements, it is more suitable to use a small- or medium-sized industrial UAV of more than 25 kg, with a certain degree of carrying capacity. We used a 50 kg vertical take-off and landing (VTOL) hybrid UAV with a flying speed of 120 km/h as the carrying platform. Figure 7 shows the appearance of this UAV. The VTOL UAV uses batteries for vertical take-off and landing, and a gasoline engine provides the cruising power. This type of design significantly reduces the demand for take-off and landing sites while ensuring long-term cruises.

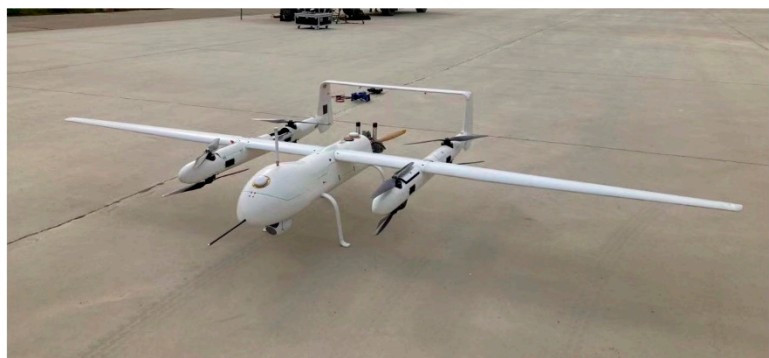

**Figure 7.** A VTOL industrial hybrid UAV weighs 50 kg, for carrying the entire real-time object search system.

## 2.2. Design of Software System

According to the functions, this section divides the software system into the following two subsystems for discussion and design: image acquisition and object detection.

### 2.2.1. Image Acquisition Subsystem

The image acquisition subsystem is mainly responsible for the camera connection check, camera control, and image transmission and storage. The workflow of this subsystem is shown in Figure 8. This subsystem focuses on the acquisition of original images and other useful information and requires efficient collaboration of various hardware. The work of each step will be explained in detail herein.

- **Camera Check**. Check the connection status between the camera and the embedded edge device. Until the device can successfully identify the camera, the subsystem will enter the next step. Otherwise, it will execute this step in a loop;
- **Image Capture**. The embedded edge device will control the camera to automatically capture images after the camera is successfully connected. The camera can capture images at its own fastest response or capture images according to a preset fixed time;
- **Read Image from Camera**. Read the currently captured image of the camera and transfer it to the embedded edge device. For the sake of efficiency, the image will be read directly from the camera's memory and then temporarily stored in the edge device's memory, without passing through low-speed links, such as the camera SD card storage;
- **Send Data to UAV**. The working status of the system is sent to the UAV and then transmitted to the GCS for real-time monitoring after being forwarded by the UAV;
- **Read Data from UAV**. Read data from the UAV's flight control computer (FCC). The FCC provides geographic coordinates (latitude and longitude), speed, altitude, and attitude (pitch, roll, and yaw) data for the edge device, and the edge device uses UAV data to match each aerial image in real-time. This step will also receive system control commands from GCS forwarded by the UAV, which can modify the camera trigger frequency, the threshold of the object detection algorithm, etc. online;
- **Write to Local Disk**. Save the original image in the memory of the embedded edge device to the local disk and store the corresponding UAV data simultaneously. This step retains all of the original data, which can be used for subsequent offline analysis;
- **Enqueue**. A queue is established in the memory of the edge device to be responsible for transmitting the data of the image acquisition subsystem to the object detection subsystem. In this step, the original image and corresponding UAV data in the memory are also put into the queue for processing by the following object detection subsystem.

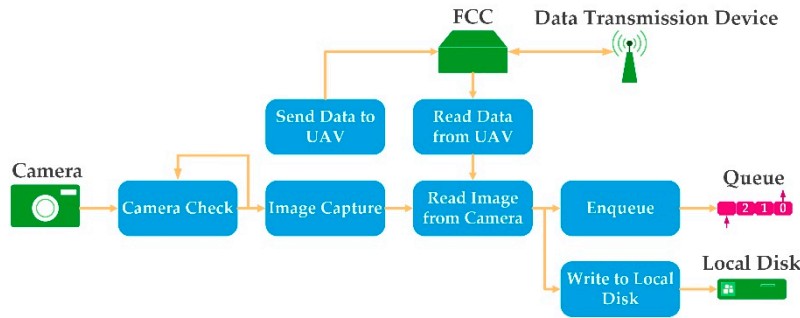

**Figure 8.** The detailed step flowchart of the image acquisition subsystem.

2.2.2. Object Detection Subsystem

The object detection subsystem is mainly responsible for image object detection, suspected object positioning, and real-time transmission to the GCS for display. The workflow of this subsystem is shown in Figure 9.

- **Dequeue.** After processing by the image acquisition subsystem, the original image and the corresponding UAV data have been temporarily stored in the memory queue. In this step, the temporarily stored data is dequeued in pairs according to the first-in-first-out principle for subsequent image object detection and suspect object positioning.
- **Object Detection.** Detect suspicious objects in the aerial image. In order to improve the object detection algorithm's inference speed and make it better applied on mobile edge devices, we have simplified the object detection algorithm and used the TensorRT to accelerate the object detection model. We will describe this part in detail later, here we are mainly concerned with how to detect aerial images with a large field of view. For the large field of view and small objects of aerial images, the object detection algorithm cannot process the entire image at once. In YOLT [39], a method for detecting small objects in satellite images is proposed, and our system refers to this method in the embedded edge device. In the inference stage, the image is first divided into $N \times N$ blocks of equal size and guarantees a certain degree of overlap in order to prevent the object from being split. The choice of $N$ depends on the resolution of the image and the size of the object that is to be searched. If the object to be searched is relatively small, a large $N$ should be selected for finer identification; otherwise, a small value can be selected for $N$ to save the hardware resources. Then, object detection is performed on each block separately. After the detection is completed, merge all of the divided small images, as shown in Figure 10, and intercept the suspected object images and temporarily store them in the memory of the embedded edge device.
- **Calculate Object's Location**. The image acquisition subsystem has recorded a geographic coordinate for each image. However, this coordinate is only the location of the UAV at the moment the camera is triggered, not the suspected object (Figure 11a). Moreover, the attitude of the UAV when the camera is triggered will also cause the tilt and rotation of the image, which increases the complexity of the problem (Figure 11b). The height and width of each image usually represent the actual distance of hundreds of meters. If a large attitude accompanies the UAV at this time, the offset of the object may even reach more than 1 km, which significantly impacts the object's positioning.

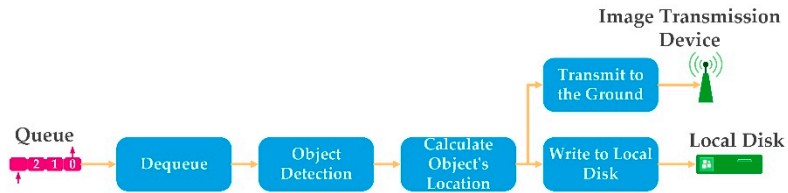

**Figure 9.** The detailed step flowchart of the object detection subsystem.

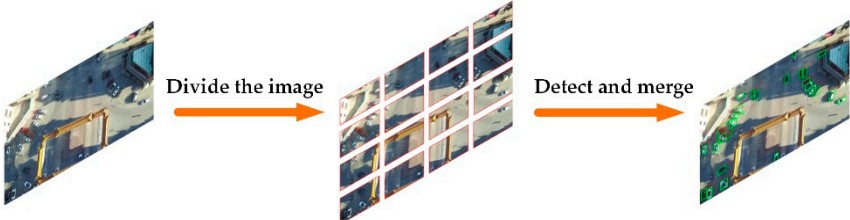

**Figure 10.** The aerial image is first divided into small images of equal size with overlap for detection and then merged after all of the detection is completed.

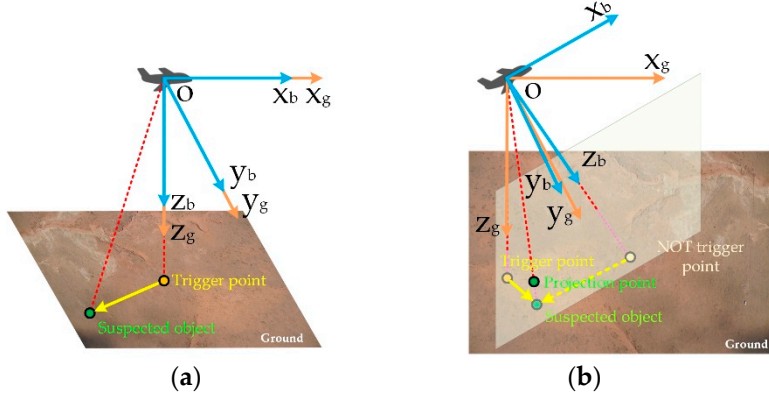

**(a)**                    **(b)**

**Figure 11.** (**a**) The situation where the pitch, roll, and yaw are all zero during the flight of the UAV (basically impossible). The recorded coordinate is the exact center of the captured image; (**b**) When the pitch, roll, and yaw are not all zero, the airframe coordinate system does not coincide with the geodetic coordinate system, and the captured image is deformed.

To this end, we combine the UAV data to map the XY coordinate of the suspected object in the image to the actual geographic coordinate. Figure 11a depicts the ideal situation, when pitch, roll, and yaw are all zero. At this time, the UAV's airframe coordinate system ($x_b$, $y_b$, $z_b$ axis) entirely coincides with the geodetic coordinate system ($x_g$, $y_g$, $z_g$ axis), and the recorded geographic coordinate is the center of the image. It is easy to calculate the coordinate of the suspected object according to the flying altitude. When the pitch, roll, and yaw are not all zero, it becomes the situation shown in Figure 11b, and the airframe coordinate system is misaligned with the geodetic coordinate system. The transformation matrix can be used to transform the points on the projection plane (white translucent area) of the airframe coordinate system to the coordinates of the geodetic coordinate system. We define the pitch angle as $\theta$, the roll angle as $\varphi$, and the yaw angle as $\psi$, the transformation matrix is defined as follows:

$$B = \begin{bmatrix} \cos\theta\cos\psi & \cos\theta\sin\psi & -\sin\theta \\ \sin\theta\sin\varphi\cos\psi - \cos\varphi\sin\psi & \sin\theta\sin\varphi\sin\psi + \cos\varphi\cos\psi & \cos\theta\sin\varphi \\ \sin\theta\cos\varphi\cos\psi + \sin\varphi\sin\psi & \sin\theta\cos\varphi\cos\psi - \sin\varphi\cos\psi & \cos\theta\cos\varphi \end{bmatrix} \quad (1)$$

Assuming the flying altitude relative to ground is $H$, for each point ($x_b$, $y_b$, $H$) on the projection plane of the airframe coordinate system, it can be transformed into the following geodetic coordinate ($x_g$, $y_g$, $z_g$):

$$\begin{bmatrix} x_g & y_g & z_g \end{bmatrix} = \begin{bmatrix} x_b & y_b & H \end{bmatrix} \cdot B \quad (2)$$

The coordinate can then be further projected to the true ground in geodetic coordinate system, and this is the work described in Figure 12. Suppose the final ground coordinate is $(x, y, z)$, it can be solved according to the following formula:

$$\begin{aligned} \frac{z_g}{H} &= \frac{x_g}{x} \\ \frac{z_g}{H} &= \frac{y_g}{y} \\ z &= H \end{aligned} \tag{3}$$

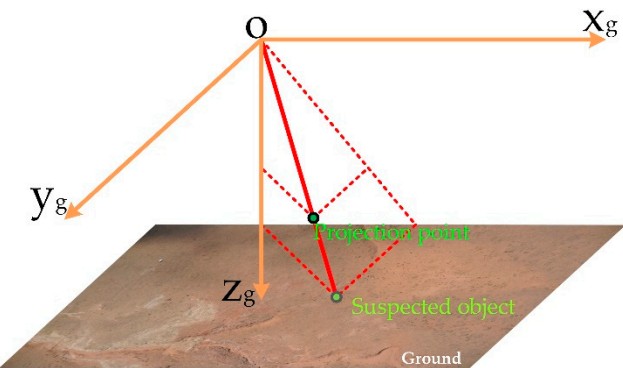

**Figure 12.** A schematic diagram of projecting the projection point to the actual position of the suspected object on the true ground.

After extracting the dependent variable, as follows:

$$\begin{aligned} x &= \frac{x_g}{z_g} \cdot H \\ y &= \frac{y_g}{z_g} \cdot H \\ z &= H \end{aligned} \tag{4}$$

- **Write to Local Disk**. Save the image and the positioning data of the suspected objects to the local disk for subsequent offline analysis.
- **Transmit to the Ground**. The suspected object images are further screened according to the image size and the score of the detection results to facilitate the real-time transmission to the GCS through the image transmission device. The image transmission device is a kind of equipment carried by the UAV for long-distance wireless transmission of images.

2.2.3. Optimization of Software System

From the above discussion of the two subsystems, it can be seen that there are many steps to go through from the acquisition of the image to the transmission of the detection results from the ground. If only the processing method of sequential execution is adopted, the system is not efficient enough. Benefitting from the parallel computing capabilities of the Jetson AGX Xavier, the two subsystems can further divide into several modules for multi-thread parallel processing in order to improve the running performance. Using this parallel pipeline work strategy, although the total processing time of a single image remains unchanged, the system's overall efficiency has been greatly improved.

The whole software system can be divided into six modules to run in parallel, as shown in Figure 13. For the image acquisition subsystem, the *Image Acquisition* module contains three steps of **Camera Check**, **Image Capture**, and **Read Image from Camera**; the *Communicate with UAV* module contains two steps of **Read Data from UAV** and **Send Data to UAV**; and the *Store Original Image* module contains two steps of **Write to Local Disk** and **Enqueue**. For the object detection subsystem, *the Object Detection* module contains two steps of **Dequeue** and **Object Detection**; the *Store Detect Results* module contains two steps of **Calculate Object's Location** and **Write to Local Disk**; and the *Results Real-time Transmit* module contains one step of **Transmit to the Ground**.

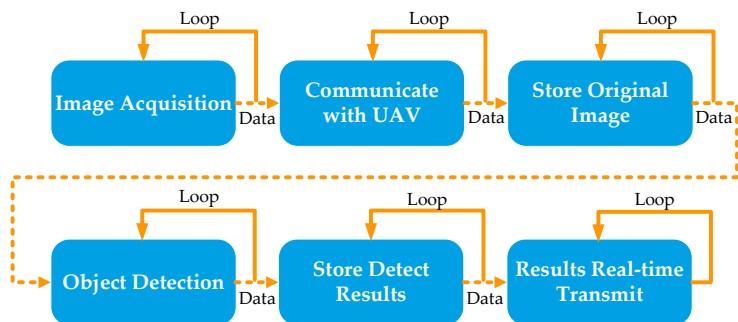

**Figure 13.** Divide the software system into modules that execute in parallel. After each module has finished running and the data is handed over to the next module, it will be executed in a loop immediately.

### 2.3. Object Detection Algorithm

#### 2.3.1. Algorithm Simplification

Since the system needs to be applied to a mobile edge device with limited computing power, the speed of the object detection algorithm is the critical bottleneck of the system's performance. Given the characteristics of aerial images, we have carried out some general simplification strategies for different commonly used algorithms in order to weigh the precision and efficiency of object detection. We simplified the algorithms from the following three aspects:

- **Remove High-level Feature Maps**. High-level feature maps have a larger receptive field. Since the objects in the aerial image are mainly medium-sized (pixel size is between $32^2$ and $96^2$, COCO metrics), a situation in which the object occupies a large area of the image will hardly appear, so we try to remove some high-level feature maps. When the feature pyramid network (FPN) [40] is used as the neck of the model, only three low-level feature maps are retained, as shown in Figure 14. We also discussed the popular YOLOv5 algorithm with a different structure. The YOLOv5 algorithm has three detection heads with a stride of 8, 16, and 32 respectively, and the head of a stride 32 (P5) is removed for experimentation. The smaller S model (YOLOv5s) was selected as the benchmark, and the simplified version of YOLOv5s are shown in Figure 15.
- **Reduce the Channels of the Intermediate Layer**. The FPN's [40] output channels and the feature map channels of classification, regression, and region proposal network [18] (RPN, if it exists) are adjusted from 256 to 128. For YOLOv5s, we directly change the "width_multiple" parameter from 0.50 to 0.30 to reduce the channels. This simplified method can effectively reduce the number of parameters of the algorithm.
- **A Lightweight Backbone**. Using a lightweight backbone also helps to reduce the model's size and increase the inference speed. We tried to replace ResNet50 [41] with ResNet18, and YOLOv5s can directly modify "depth_multiple" to control the depth of the model. This paper has also performed an experiment with replacing CPSDarkNet53 in YOLOv5s with MobileNetV2 [42] directly.

#### 2.3.2. TensorRT

It is often inefficient to use the deep learning framework for model deployment in actual applications directly. Instead, deploying object detection models trained on mainstream frameworks on the NVIDIA GPU through TensorRT can significantly improve inference speed, often at least one time faster than the original. TensorRT is a deep learning inference optimizer that transforms the trained model through a series of optimization techniques in order to enable it to run with higher performance on the NVIDIA GPU of a specific platform. Specifically, TensorRT has the following two main optimization strategies:

- **Lower Data Precision**. For most of the deep learning frameworks in training, the tensors in the network use 32-bit floating-point precision (FP32). Since the backpropa-

gation is not required during the inference process, TensorRT supports FP16 and INT8 quantization to reduce data precision appropriately. Table 3 lists the value ranges of the different data types. Lower data precision represents lower memory usage and smaller model size.

- **Reconstruction and Optimization of Network**. For NVIDIA GPUs, TensorRT dramatically reduces the number of compute unified device architecture (CUDA) cores by merging layers, horizontally or vertically, as depicted in Figure 16. Horizontal merging can integrate the convolution, bias, and activation layers into one structure, while vertical merging can integrate the layers with the same structure but with different weights into a wider layer.

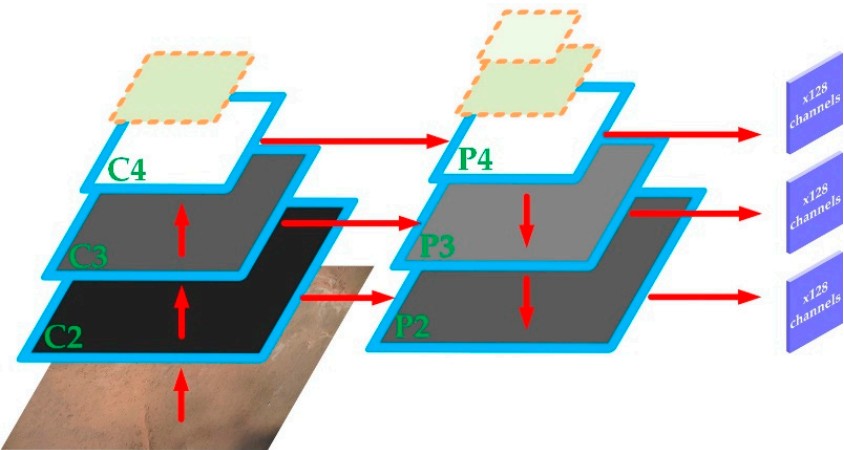

**Figure 14.** Retain three feature maps in FPN and change the output channels to 128. The RPN, classification, and regression feature map channels are also changed.

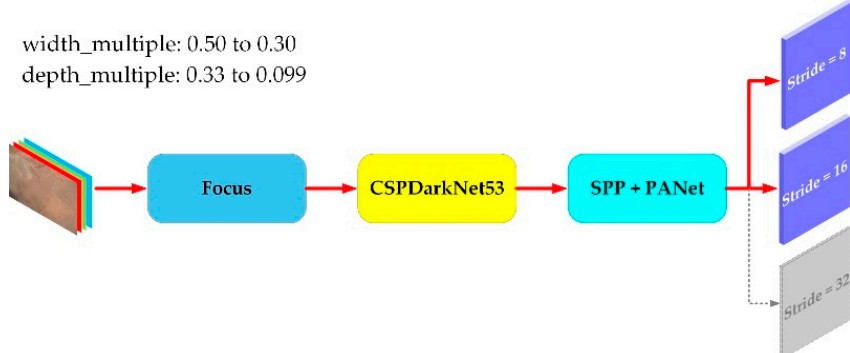

**Figure 15.** Schematic diagram of YOLOv5s. The simplification of YOLOv5s mainly includes removing the head whose stride is 32 (P5), reducing the parameters of "width_multiple" and "depth_multiple".

**Table 3.** Value ranges of different data types.

| Data Type | Number of Bytes | Dynamic Range |
| --- | --- | --- |
| FP32 | 4 | $-3.4 \times 10^{38}$~$3.4 \times 10^{38}$ |
| FP16 | 2 | $-65{,}504$~$65{,}504$ |
| INT8 | 1 | $-128$~$127$ |

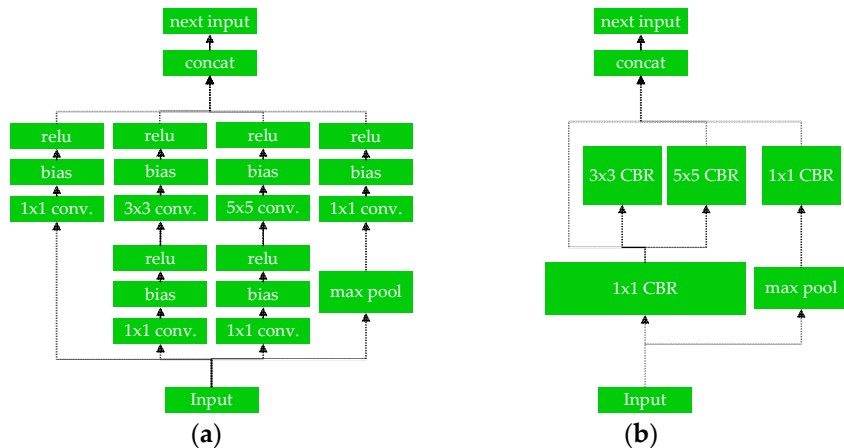

**Figure 16.** (**a**) The original network structure; (**b**) The network structure after reconstruction and optimization by TensorRT.

### 2.3.3. Data Augmentation

Some data augmentation methods that do not affect the inference speed are used to expand the existing training dataset in order to enhance the adaptability of the actual scene, according to the features of the aerial images that are collected by the UAV. As listed in Figure 17, the data augmentation methods adopted are as follows:

- **Blur**. Small objects in high-resolution aerial images collected by UAV will be blurred to some extent.
- **Ghosting**. Since the camera is mounted on a UAV that is flying at high speed, some of the captured images may show ghosting.
- **Lighting**. Depending on the date and weather when the task is being performed, the lighting conditions will vary.

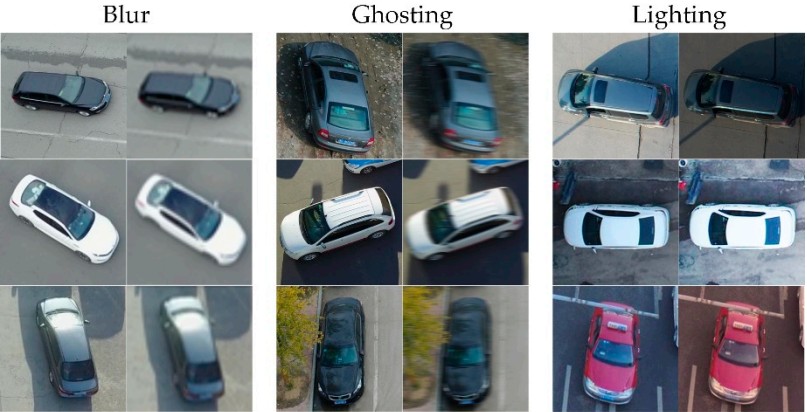

**Figure 17.** Data augmentation for aerial images.

## 3. Experiments

We trained and validated the simplified object detection algorithms for vehicle detection in aerial images and applied the entire wide-area and real-time object search system to a practical engineering task.

### 3.1. Simplification of Object Detection Algorithms

#### 3.1.1. Dataset

We chose VisDrone2021-DET aerial images as the dataset. The VisDrone [43] dataset is collected and released by researchers in the Lab of Machine Learning and Data Mining, Tianjin University, China. All of the data are captured by drones at different locations and

heights and consists of more than 400 video clips with 265,228 frames and 10,209 static images. VisDrone2021-DET is a dataset used by the VisDrone team for the image object detection challenge. The challenge used all 10,209 static images, of which 6471 were used for training, 548 were used for validating, and 3190 were used for testing. Figure 18 shows some of the images in the dataset. There are ten types of objects in the dataset, including the following: pedestrian, person, car, van, bus, truck, motor, bicycle, awning-tricycle, and tricycle, and our training was only conducted for the three categories of car, van, and truck. In addition to the original dataset, we also used the methods described in the previous chapter for data augmentation.

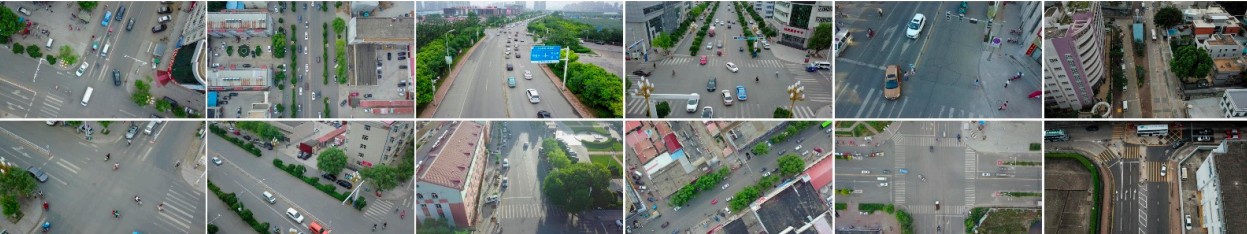

**Figure 18.** Some images of the VisDrone2021-DET dataset.

### 3.1.2. Benchmarks

We selected six common object detection algorithms as the benchmarks, including YOLOv5s, and Faster RCNN, Cascade RCNN, RetinaNet, FCOS, ATSS whose backbone is ResNet50 and neck is FPN.

### 3.1.3. Configuration

The training of all of our models was performed on a 2080 Ti graphics card with 11 GB of video memory. In order to facilitate the comparison of the results, we uniformly did not use the pre-trained model during training. The training epochs of the models were set to 20, and the batch size was eight. For YOLOv5s, the image size was 800 during training and testing, and the OneCycleLR learning rate policy was used, in which lr0 was 0.01 and lrf was 0.2, warmup was performed in the first three epochs. For other models, the image sizes were all set to $800 \times 600$, the initial learning rate was 0.005, the momentum was 0.9, and the weight decay was 0.0001. The learning rate warmup was performed in the first 500 iterations, and then the training was maintained at a fixed learning rate. When learning 14–18 epochs, the learning rate will be 1/10 of the original value. In the last two epochs, it was reduced to 1/100 of the original learning rate.

### 3.1.4. Metrics

We used the following six metrics to evaluate the performance of the model: mean average precision (mAP), mAP decline relative to the benchmark, trainable parameters, floating point operations (FLOPs), frames per second on the Jetson AGX Xavier embedded edge device (FPS), and FPS improving rate.

### 3.1.5. Results

The results of our experiments are shown in Table 4. For the algorithms using the ResNet-FPN structure, removing the high-level feature maps and reducing the number of channels in the intermediate layers had little effect on the model's mAP, and the mAP decline was within 1%. However, after replacing ResNet50 with ResNet18, the mAP loss was generally higher, reaching a maximum of 3.6%. The three simplification methods all improved the inference speed to varying degrees. For the YOLOv5s algorithm, removing the detection head of P5 basically did not affect the mAP, and the speed improvement was also very slight. Although reducing the "width_multiple" and "depth_multiple" parameters improved the detect FPS, they also sacrifice some mAP. MobileNetV2 replacement caused a large loss of mAP, which is not a good idea.

**Table 4.** Comparison results of various simplified methods of object detection algorithms. In this table, R50/R18 stands for ResNet50/ResNet18, and S1 represents the simplification of removing high-level feature maps on the model, only retaining the three low-level feature maps of the FPN. S2 represents the simplification of changing the number of channels of all intermediate layers, from 256 to 128. For the structure of YOLOv5s, Y1 means that the detection head with a stride of 32 (P5) is not used for detection, Y2 represents reducing the "width_multiple" from 0.50 of YOLOv5s to 0.30, and Y3 represents reducing the "depth_multiple" from 0.33 of YOLOv5s to 0.099. MobileNetV2 means that the backbone of YOLOv5s is replaced. The input size of YOLOv5s is 800, and the size of other models are 800 × 600. The calculation of FPS considers the time consumed by the preprocessing and postprocessing procedures.

| Model | mAP | mAP Decline | Parameters | FLOPs | FPS |
|---|---|---|---|---|---|
| Faster-RCNN-R50-FPN | 43.3% | / | 41.12 M | 104.59 G | 4.06 |
| Faster-RCNN-R50-FPN (S1) | 42.8% | 0.5% | 40.01 M | 103.70 G | 4.17 |
| Faster-RCNN-R50-FPN (S2) | 43.0% | 0.3% | 31.99 M | 60.89 G | 5.53 |
| Faster-RCNN-R18-FPN | 41.2% | 2.1% | 28.12 M | 79.66 G | 5.18 |
| Cascade-RCNN-R50-FPN | 45.4% | / | 68.93 M | 132.39 G | 3.01 |
| Cascade-RCNN-R50-FPN (S1) | 45.2% | 0.2% | 67.81 M | 131.50 G | 3.09 |
| Cascade-RCNN-R50-FPN (S2) | 45.2% | 0.2% | 46.95 M | 75.85 G | 4.16 |
| Cascade-RCNN-R18-FPN | 44.4% | 1.0% | 55.93 M | 107.46 G | 3.51 |
| RetinaNet-R50-FPN | 35.0% | / | 36.10 M | 96.34 G | 4.66 |
| RetinaNet-R50-FPN (S1) | 34.9% | 0.1% | 32.04 M | 95.50 G | 4.93 |
| RetinaNet-R50-FPN (S2) | 34.2% | 0.8% | 27.92 M | 54.71 G | 5.89 |
| RetinaNet-R18-FPN | 32.1% | 2.9% | 19.61 M | 72.42 G | 6.12 |
| FCOS-R50-FPN | 41.1% | / | 31.84 M | 92.69 G | 4.88 |
| FCOS-R50-FPN (S1) | 40.7% | 0.4% | 30.13 M | 91.56 G | 5.17 |
| FCOS-R50-FPN (S2) | 40.2% | 0.9% | 25.62 M | 51.75 G | 6.39 |
| FCOS-R18-FPN | 37.5% | 3.6% | 18.93 M | 71.47 G | 6.49 |
| ATSS-R50-FPN | 46.2% | / | 31.89 M | 94.92 G | 4.73 |
| ATSS-R50-FPN (S1) | 46.0% | 0.2% | 30.18 M | 93.79 G | 4.97 |
| ATSS-R50-FPN (S2) | 45.7% | 0.5% | 25.67 M | 53.98 G | 5.96 |
| ATSS-R18-FPN | 43.2% | 3.0% | 18.94 M | 71.47 G | 6.16 |
| YOLOv5s | 44.0% | / | 7.05 M | 12.75 G | 11.39 |
| YOLOv5s (Y1) | 43.9% | 0.1% | 5.27 M | 11.64 G | 11.66 |
| YOLOv5s (Y2) | 40.7% | 3.3% | 2.69 M | 5.21 G | 13.76 |
| YOLOv5s (Y3) | 42.8% | 1.2% | 6.64 M | 11.11 G | 13.18 |
| YOLOv5s (MobileNetV2) | 35.2% | 8.8% | 4.54 M | 7.66 G | 13.43 |

We integrated all of the simplification strategies in order to perform a comprehensive experiment and the results are shown in Table 5. After our simplification, most of the algorithms can achieve an efficiency improvement of more than 90%, with a slight loss of mAP. It can be seen from the table that the speed improvement of simplifying the YOLOv5s algorithm is relatively small, which may be due to YOLOv5 itself having already integrated so many optimization strategies.

### 3.2. TensorRT Acceleration Experiments

TensorRT will reconstruct the network structure by default. In addition, it also has a quantization strategy, which can convert the original model into two low-precision models of FP16 and INT8. FP16 and INT8 have only two and one bytes, respectively, and some information may be lost when they are used to represent a 4-byte FP32 value. The loss caused by the conversion to different precision models is shown in Table 6.

**Table 5.** The final result of object detection algorithm simplification. ALL means that the algorithm uses all of the simplification strategies mentioned in Table 3, and Y1 + Y2 + Y3 means that only the simplification strategy of a specific sign is used.

| Model | mAP | mAP Decline | Parameters | FLOPs | FPS | Improving Rate |
|---|---|---|---|---|---|---|
| Faster-RCNN-R50-FPN | 43.3% | / | 41.12 M | 104.59 G | 4.06 | |
| Faster-RCNN (ALL) | 40.3% | 3.0% | 19.15 M | 37.15 G | 8.12 | 100.0% |
| Cascade-RCNN-R50-FPN | 45.4% | / | 68.93 M | 132.39 G | 3.01 | |
| Cascade-RCNN (ALL) | 43.5% | 1.9% | 34.11 M | 52.11 G | 5.75 | 91.0% |
| RetinaNet-R50-FPN | 35.0% | / | 36.10 M | 96.34 G | 4.66 | |
| RetinaNet (ALL) | 31.3% | 3.7% | 12.89 M | 31.43 G | 9.53 | 104.5% |
| FCOS-R50-FPN | 41.1% | / | 31.84 M | 92.69 G | 4.88 | |
| FCOS (ALL) | 36.4% | 4.7% | 12.69 M | 30.92 G | 9.80 | 100.8% |
| ATSS-R50-FPN | 46.2% | / | 31.89 M | 94.92 G | 4.73 | |
| ATSS (ALL) | 42.6% | 3.6% | 12.70 M | 30.92 G | 9.33 | 97.2% |
| YOLOv5s | 44.0% | / | 7.05 M | 12.75 G | 11.39 | |
| YOLOv5s (Y1 + Y2 + Y3) | 39.9% | 4.1% | 1.85 M | 4.15 G | 15.71 | 37.9% |

**Table 6.** Performance comparison after converting the original model to FP16 or INT8 model using TensorRT. Cali. indicates that the model was calibrated before INT8 quantization, and engine represents a model file is suffixed with engine.

| Model | Precision | mAP | mAP Decline | Model Volume |
|---|---|---|---|---|
| Faster-RCNN (ALL) | Original model | 40.3% | / | 120.4 MB |
| | FP16 | 38.9% | 1.4% | 37.7 MB |
| | INT8 | 37.4% | 2.9% | 24.3 MB |
| | INT8 (Cali.) | 38.7% | 1.6% | 29.5 MB |
| Cascade-RCNN (ALL) | Original model | 43.5% | / | 240.0 MB |
| | FP16 | 40.7% | 2.8% | 82.8 MB |
| | INT8 | 39.4% | 4.1% | 47.4 MB |
| | INT8 (Cali.) | 40.5% | 3.0% | 52.7 MB |
| RetinaNet (ALL) | Original model | 31.3% | / | 70.3 MB |
| | FP16 | 31.1% | 0.2% | 27.9 MB |
| | INT8 | 8.1% | 23.2% | 21.4 MB |
| | INT8 (Cali.) | 31.0% | 0.3% | 21.7 MB |
| FCOS (ALL) | Original model | 36.4% | / | 68.8 MB |
| | FP16 | 36.3% | 0.1% | 28.2 MB |
| | INT8 | 4.5% | 31.9% | 22.2 MB |
| | INT8 (Cali.) | 35.8% | 0.6% | 21.7 MB |
| ATSS (ALL) | Original model | 42.6% | / | 68.8 MB |
| | FP16 | 40.9% | 1.7% | 27.7 MB |
| | INT8 | 37.6% | 5.0% | 21.7 MB |
| | INT8 (Cali.) | 40.7% | 1.9% | 21.3 MB |
| YOLOv5s (Y2 + Y3) | Original model | 39.8% | / | 5.6 MB |
| | FP16 | 38.5% | 1.3% | 6.4 MB (engine) |
| | INT8 (Cali.) | 38.3% | 1.5% | 1.4 MB (engine) |

It can be seen from the table that, after converting to the FP16 model, various algorithms have produced different degrees of mAP decline, but they were all within an acceptable range. Cascade RCNN, which had the least drop in mAP during model simplification, had the most drop during TensorRT deployment. Converting to the INT8 model requires the assistance of a calibration dataset. We performed a set of experiments without the calibration dataset and it turns out that the model loses too much mAP, which is pre-

dictable since INT8 had only 256 value representation ranges. More seriously, two of the models (RetinaNet and FCOS) diverged after INT8 quantization (marked in green font in Table 6). TensorRT takes this into account and provides a fully automatic calibration process for optimization in order to minimize the performance loss after INT8 quantization, which only requires some images in a similar style to the training dataset. We randomly selected 1000 images from the VisDrone2021-DET testing dataset as the calibration dataset. After calibration, the performance of the INT8 model was greatly improved. Most of the INT8 models had a mAP difference of only 0.2% from the FP16 model, but they were smaller and faster. When converting the YOLOv5s model, we retained the detection head of P5, which had little effect on precision and efficiency. The FP16 and INT8 model generated by YOLOv5s was a file with a engine suffix, but this did not affect the deployment effect.

We adopted the INT8 model for subsequent experiments. On the Jetson AGX Xavier edge device, whether TensorRT was used for deployment or not, and the comparison of the inference speeds of the different object detection algorithms are shown in Table 7. It can be seen that the inference speed of object detection models has been greatly improved after TensorRT deployment, reaching a maximum improvement rate of 140%, which is very critical for mobile edge devices with limited performance.

**Table 7.** Comparison of inference speed with and without TensorRT deployment (INT8).

| Model | FPS (Original) | FPS (TensorRT) | Improvement Rate |
|---|---|---|---|
| Faster-RCNN (ALL) | 8.12 | 18.72 | 130.5% |
| Cascade-RCNN (ALL) | 5.75 | 13.85 | 140.8% |
| RetinaNet (ALL) | 9.53 | 20.20 | 111.9% |
| FCOS (ALL) | 9.80 | 20.46 | 108.7% |
| ATSS (ALL) | 9.33 | 19.91 | 113.3% |
| YOLOv5s (Y2 + Y3) | 15.48 | 27.20 | 75.7% |

Table 8 summarizes all of the previous experiments. After model simplification and TensorRT deployment, most of the models sacrifice only about 5% of mAP, but in exchange for at least 400% of the operating efficiency of the benchmark model.

**Table 8.** Comparisons of the performance of the final model used on the edge device and the benchmark model.

| Benchmark Model | mAP | mAP Final | mAP Decline | FPS | FPS Final | Improvement Rate |
|---|---|---|---|---|---|---|
| Faster-RCNN-R50-FPN | 43.3% | 38.7% | **4.6%** | 4.06 | 18.72 | **361.0%** |
| Cascade-RCNN-R50-FPN | 45.4% | 40.5% | **4.9%** | 3.01 | 13.85 | **360.1%** |
| RetinaNet-R50-FPN | 35.0% | 31.0% | **4.0%** | 4.66 | 20.20 | **333.4%** |
| FCOS-R50-FPN | 41.1% | 35.8% | **5.3%** | 4.88 | 20.46 | **319.2%** |
| ATSS-R50-FPN | 46.2% | 40.7% | **5.5%** | 4.73 | 19.91 | **320.9%** |
| YOLOv5s | 44.0% | 38.3% | **5.3%** | 11.39 | 27.20 | **138.8%** |

*3.3. Practical Application of the System*

3.3.1. Task Analysis

Our target was to conduct a regular inspection task in a grazing area of the prairie with a range of about $7 \times 20$ km$^2$ in order to check for suspicious vehicles. We selected Sony A7R2 with a 55 mm prime lens as the image capture device, and the installation of this set of devices on the UAV is shown in Figure 19a. When flying at different altitudes relative to the ground, the coverage of a single acquisition is shown in Table 9.

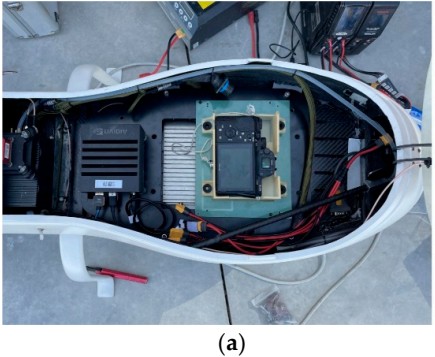
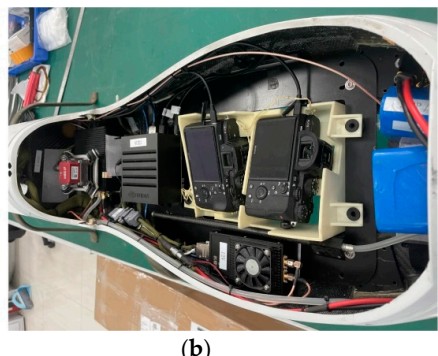

(**a**)　　　　　　　　　　　　　　　　(**b**)

**Figure 19.** (**a**) Install a single camera with the angle of view facing directly below the UAV; (**b**) Install dual cameras, and the cameras deviate from the UAV by 15° to the left or right, respectively.

**Table 9.** The coverage of a single acquisition, using a Sony A7R2 camera with a 55 mm prime lens.

| Fly Relative Altitude | Width Coverage | Height Coverage | Resolution Per Pixel |
|---|---|---|---|
| 300 m | 195.818 m | 130.909 m | 0.025 m |
| 400 m | 261.091 m | 174.545 m | 0.033 m |
| 500 m | 326.364 m | 218.182 m | 0.041 m |
| 600 m | 391.636 m | 261.818 m | 0.049 m |
| 700 m | 456.909 m | 305.455 m | 0.057 m |
| 800 m | 522.182 m | 349.091 m | 0.066 m |
| 900 m | 587.455 m | 392.727 m | 0.074 m |
| 1000 m | 652.727 m | 436.364 m | 0.082 m |

We comprehensively considered the efficiency and effect and chose a flight altitude of 800 m relative to the ground for scanning. Figure 20 is a schematic diagram of a scanning route. With a 20% image side overlap rate, the distance between the two adjacent routes was calculated to be about 417 m. In the flying direction of the UAV, we considered 40% overlap, and the interval was 209 m. According to the UAV's flying speed (120 km/h, 33.3 m/s), it can be calculated that the capture interval between the two images should be less than about 6.27 s.

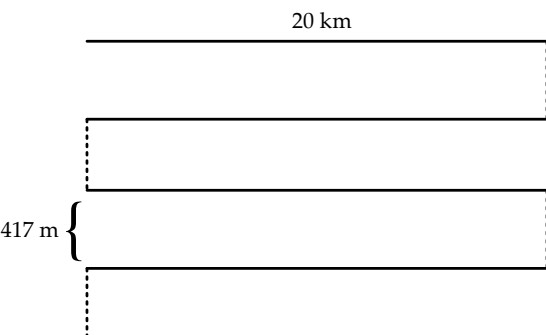

**Figure 20.** Schematic diagram of the scanning route.

### 3.3.2. System Performance

We already know that the entire system was executed in parallel by six modules. In the task of vehicle search in a prairie grazing area equipped with a single camera, the actual execution period of each module is shown in Table 10. We chose simplified Faster RCNN as the object detection algorithm of the system and selected *N* = 5 for each aerial image collected by the camera. That is, each aerial image needed to be detected 25 times in total. As we can see from the table, compared with sequential processing, modular parallel processing provided a speed increase of more than 100%. The average period of

parallel execution was only 3.8 s, which is equal to the time consumption of the slowest Image Acquisition module in the system. Modular processing also brings good scalability and versatility to the system. In other practical applications, different cameras, different object detection algorithms, and different $N$ values can be flexibly selected to adapt to different scenarios.

**Table 10.** Execution time of each module of the system.

| Module | Average Execution Period |
|---|---|
| Image Acquisition (Camera = 1) | 3.8 s |
| Communicate with UAV | Ignored |
| Store Original Image | 0.9 s |
| Object Detection ($N = 5$) | 2.4 s |
| Store Detect Results | 1.1 s |
| Results Real-time Transmit | Ignored |
| Average sequential execution period | 8.2 s |
| Average parallel execution period | 3.8 s |

### 3.3.3. Equipped with Dual Cameras

The system further improved the working efficiency by using two cameras. The installation method of the dual cameras is shown in Figure 19b, and the principle of image acquisition is shown in Figure 21. The system captures two images at a time while ensuring that the images from the two cameras partially overlap.

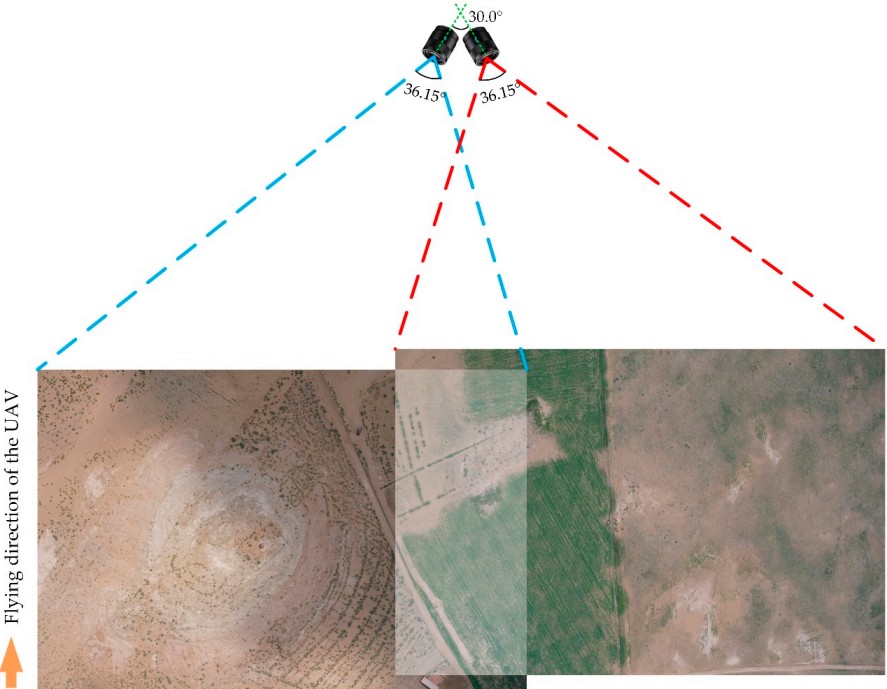

**Figure 21.** Images of two cameras with a certain degree of overlap.

When the UAV was equipped with dual cameras, the coverage of a single acquisition is shown in Table 11. As can be seen from the table, the width coverage was greatly improved. The tilt of the camera stretched the image, so the height coverage was calculated at the lowest value. After using two cameras, the resolution per pixel was reduced slightly at the same flying altitude, which lost some performance. Table 12 displays the actual execution period of each module. Since the two cameras can work in parallel, the bottleneck of the system at this time becomes the efficiency of the Object Detection module, which was

about 0.8 s longer than that of a single camera. Although the average acquisition time increased, it significantly increased the coverage area of one acquisition, so the system's overall efficiency was still higher.

**Table 11.** The coverage of a single acquisition by using two cameras.

| Fly Relative Altitude | Width Coverage | Height Coverage | Resolution Per Pixel |
|---|---|---|---|
| 300 m | 410.451 m | 130.909 m | 0.029 m |
| 400 m | 547.268 m | 174.545 m | 0.039 m |
| 500 m | 684.086 m | 218.182 m | 0.049 m |
| 600 m | 820.902 m | 261.818 m | 0.059 m |
| 700 m | 957.719 m | 305.455 m | 0.069 m |
| 800 m | 1094.537 m | 349.091 m | 0.078 m |
| 900 m | 1231.355 m | 392.727 m | 0.088 m |
| 1000 m | 1368.170 m | 436.364 m | 0.098 m |

**Table 12.** Execution cycle of each module of the dual camera system.

| Module | Average Execution Period |
|---|---|
| Image Acquisition (Camera = 2) | 4.0 s |
| Communicate with UAV | Ignored |
| Store Original Image | 1.4 s |
| Object Detection ($N = 5$) | 4.6 s |
| Store Detect Results | 1.7 s |
| Results Real-time Transmit | Ignored |
| Average sequential execution period | 11.7 s |
| Average parallel execution period | 4.6 s |

Figure 22 displays some of the images containing vehicles that were successfully detected. Since a single image can reach a volume size of about 30 MB, the capacity of the image transmission equipment is difficult to meet the complete transmission requirements. Therefore, the system will only transmit the full image after compression, and together with the corresponding suspected object images and the positioning to the GCS. After post-processing on the ground, the results will be displayed on the computer screen in real-time.

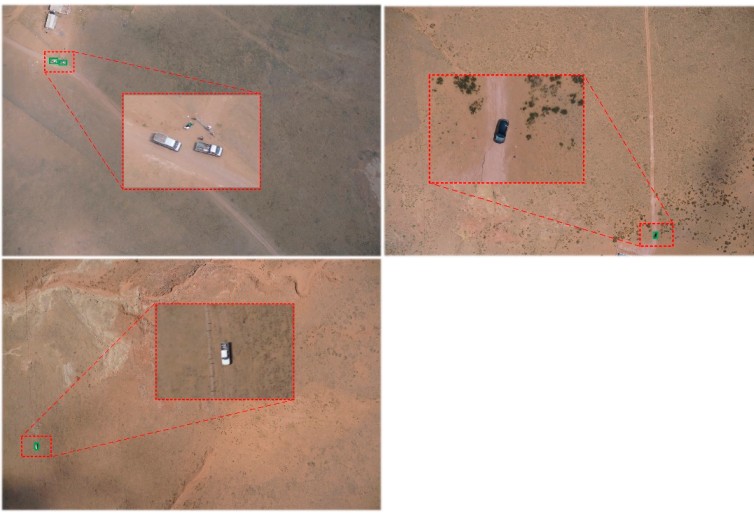

**Figure 22.** Images containing vehicles that were successfully detected.

## 4. Conclusions

This paper focuses on the real-time object search problem in a wide area. We selected the NVIDIA Jetson AGX Xavier edge device as the computing and control unit, and the high-resolution camera as the image acquisition device, to design a wide-area and real-time object search system of UAVs. The system considers both issues of real-time detection and area scanning efficiency in wide-area object search, which greatly reduces the cost of performing related tasks compared to other existing methods. Most of the hardware used in the system are shelf products, which do not need to be specially customized so that the entire system can be easily implemented.

The software part of the system is divided into the image acquisition subsystem and objection detection subsystem, which are designed respectively, and the realization and optimization scheme of each step in the subsystem are explained in detail. At the same time, the parallel multi-threading method is adopted in order to modularize the system so that the system's performance more than doubled. In this paper's discussion of the vehicle detection task, the system's execution period was reduced from 8.2 s to 3.8 s when a single camera was mounted on the UAV based on the parallel design of the software. The execution period was also compressed from 11.7 s to 4.6 s in the expansion scheme with dual cameras. The end result satisfies the time constraints calculated according to the UAV flight altitude and camera parameters in both cases. That is, the capture interval between the two images should be less than about 6.27 s. However, in the case of sequential execution, the system cannot operate normally.

For the most time-consuming object detection process, we adopted a variety of simplification strategies for the algorithms and used data augmentation on the training dataset in order to better adapt to the UAV aerial photography scene. We also adopted the TensorRT to optimize the object detection model, which significantly speeds up the detection speed and better applies to the embedded edge device with limited performance. After simplifying the algorithms and deploying the vehicle detection models on the edge device using the TensorRT tool, the detection speed of most of the models increased to 400% of the original, with only a loss of about 5% of the precision index, which solved the biggest limiting bottleneck of the system that the cost of detection time is too large.

The system design considers the scalability and versatility as much as possible, so different software and hardware modules can be flexibly selected according to different application scenarios, which provides a new idea for engineering applications in related fields.

**Author Contributions:** Conceptualization, X.L. and B.H. (Boyong He); methodology, X.L. and B.H. (Boyong He); software, X.L.; validation, K.D. and W.G.; investigation, K.D. and B.H. (Bo Huang); writing—original draft preparation, X.L.; writing—review and editing, X.L. and L.W.; supervision, L.W. All authors have read and agreed to the published version of the manuscript.

**Funding:** This research received no external funding.

**Institutional Review Board Statement:** Not applicable.

**Informed Consent Statement:** Not applicable.

**Data Availability Statement:** The data presented in this study are available upon request from the corresponding author.

**Acknowledgments:** The authors would like to thank the anonymous reviewers for their valuable comments and helpful suggestions.

**Conflicts of Interest:** The authors declare no conflict of interest.

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
