# Peer review of "Wide-Area and Real-Time Object Search System of UAV"

_remotesensing, doi:10.3390/rs14051234_

Round 1
Reviewer 1 Report
The authors present an IA-based methodology for real-time object detection in wide areas, exploiting high-performance deep learning inference instruments like Nvidia TensorRT and the Nvidia Jetson platform. The topic is of great interest for the scientific community. The paper is well written, and it describes thoroughly the research phases and the development of the strategy. Experimental results and performance analysis with many metrics show the feasibility of the approach and establish a possible new framework for UAV-based surveillance applications. I do not have particular observations, and would recommend publication of the paper in its current shape.
Reviewer 2 Report
The theme of the work is very interesting.
Comments:
- The authors are asked to compare the proposed method with other optimization and search methods to show the advantages of the proposed one.
- A lack of processor information is noticed. the authors should detail this part explaining the sampling frequency applied to compare the applied processor with other ones.
- With respect to low cost, In the paper, there is no clear explanation of the low-cost achievement applying the proposed technique in comparison with other works.
- many research works have been reviewed in the introduction section. But unfortunately, no comparison has been carried out with the suggested work.
Reviewer 3 Report
This study investigates real-time object search problem in a wide area using UAV. NVIDIA Jetson AGX Xavier device was used in the study where the software part of the system was divided into image acquisition system and objection detection system. The introduction covers good literature review of the topic and the results are looking promising. Thus the reviewer recommend the paper to be published after revising the manuscript according to the following comment.
The conclusion is too weak and must be strengthened. The authors need to add actual figures (such as the results from the experiment) to improve this section.
Round 2
Reviewer 2 Report
The authors respond fairly for the comments. I recommend the publication of the article.